# Proactive Risk Assessment through Failure Mode and Effect Analysis (FMEA) for Perioperative Management Model of Oral Anticoagulant Therapy: A Pilot Project

**DOI:** 10.3390/ijerph192416430

**Published:** 2022-12-07

**Authors:** Fausta Micheletta, Michela Ferrara, Giuseppe Bertozzi, Gianpietro Volonnino, Maria Nasso, Raffaele La Russa

**Affiliations:** 1Nuova Itor, Clinica accreditata, 00158 Rome, Italy; 2Department of Anatomical, Histological, Forensic Medicine and Orthopedic Sciences, Sapienza University of Rome, 00161 Rome, Italy; 3Department of Clinical and Experimental Medicine, Section of Forensic Pathology, University of Foggia, 71122 Foggia, Italy

**Keywords:** anticoagulant therapy, Failure Mode and Effect Analysis, elective surgery

## Abstract

Introduction: Correct perioperative management of anticoagulant therapy is essential to prevent thromboembolic events and reduce the risk of bleeding. The lack of universally accepted guidelines makes perioperative anticoagulant therapy management difficult. The present study aims to identify the perioperative risks of oral anticoagulant therapy and to reduce adverse events through Failure Mode and Effect Analysis (FMEA). Materials and Methods: A multidisciplinary working group was set up, and four main phases of the process were identified. Each of these phases was divided into micro-activities to identify the related possible failure modes and their potential consequences. The Risk Priority Number was calculated for each failure mode. Results and Discussion: Seventeen failure modes were identified in the entire perioperative period; those with a higher priority of intervention concern the incorrect timing between therapy suspension and surgery, and the incorrect assessment of the bleeding risk related to the invasive procedure. Conclusion: The FMEA method can help identify anticoagulant therapy perioperative failures and implement the management and patient safety of surgical procedures.

## 1. Introduction

Every year, 10–15% of individuals taking oral anticoagulation (OA) undergo invasive surgical procedures [1]; most of these are patients with a history of atrial fibrillation (AF), venous thromboembolism, or prosthetic heart valves. It is estimated that about one in six patients with atrial fibrillation, which is the prevalent clinical indication for long-term anticoagulant therapy, requires an annual interruption of treatment before an elective surgical procedure [2,3]. Moreover, the use of anticoagulant therapy is spreading thanks to the availability of so-called direct oral anticoagulants (DOAC), as they directly and exclusively inhibit IIa or Xa coagulation factors. DOACs (Apixaban, Dabigatran Etexilate, Edoxaban, and Rivaroxaban) have gradually replaced the Vitamin K antagonists (VKA—warfarin, acenocoumarol) which, for years, represented the gold standard therapy for prevention and treatment of thrombotic events, partly because they require frequent laboratory monitoring of INR. 

OA therapy significantly affects hemostatic processes, and the management of such patients is frequently complex, as it is necessary to balance the thromboembolic and bleeding risk during perioperative surgery. Thus, it is mandatory to evaluate the various phases that concern the perioperative period, in particular: correct modality and timing of suspension of the anticoagulant drug; correct method and timing of introduction of the so-called “bridging therapy” when indicated; and correct post-operative monitoring and timing of resumption of oral anticoagulant therapy.

Despite the published recommendations by scientific societies [4,5,6,7,8], the lack of evidence from prospective studies makes it difficult to handle surgical patients. This is significant especially in the elderly, who generally undergo polydrug therapy are affected by pathological conditions such as heart failure, impaired renal function, or dehydration. In addition, it is essential to stratify the risk according to the type of scheduled surgical and anesthesiologic procedures [8]. All these aspects expose these processes to a high risk of adverse events, to be analyzed by health care systems, as nearly 50% of them are estimated to be preventable [9] through a risk management policy, which includes proactive and reactive tools. 

Among them, the Failure Mode and Effect Analysis (FMEA) methodology represents a proactive risk management tool aimed to identify possible errors within a clinical-care process and their possible consequences, to reduce adverse events and make the process safer. This tool was used for the first time in the 1940s in the military field, to analyze possible errors and consequences on the outcome of the missions and on the safety of the personnel and equipment involved [10]. Similarly, healthcare systems as complex organizations, being at high risk of errors and adverse events, need control measures to guarantee the safety of the care provided. The most conventionally used method of analysis is Root Cause Analysis (RCA) oriented towards the research to identify the main causes of adverse events that have already occurred, thus supporting the recommendation of corrective measures. In contrast, Failure Mode and Effect Analysis (FMEA) is a proactive and future-oriented method for identifying potential failures before they occur [11]. To ensure the effectiveness of this methodology, generally, the risk assessment is guided by a multidisciplinary group of experts with different experiences and technical skills [12]. The FMEA methodology requires some sequential steps which consist of analyzing the clinical process and deconstructing it into macro-activities; within each macro-activity, the relative micro-activities and all possible failure modes will be identified. Secondly, an analysis of the individual failure modes is conducted to recognize all potential consequences. Thereafter, the so-called Risk Priority Number (RPN) is calculated for each failure mode, to prioritize them, according to the probability of occurrence, detectability of the error, and the severity of consequences. Thus, the multidisciplinary team can build a master list of priorities and define a specific corrective action plan with a schedule based on the priorities identified. Finally, monitoring actions are carried out to verify the effectiveness of the implemented measures.

In this context, the purpose of the present study is to define an appropriate and safe perioperative management model of oral anticoagulant therapy in order to reduce hemorrhagic and thrombotic complications in the perioperative period. To develop this model, the authors used the FMEA methodology with the following specific objectives:(a)To identify the error modes and the related causes responsible for both thrombotic and hemorrhagic adverse events;(b)To plan specific corrective actions in the perioperative management of oral anticoagulant therapy;(c)To provide operational procedures and standardized management tools to the healthcare personnel involved in the process.

## 2. Materials and Methods

A multidisciplinary working group including various professional figures was set up. The members included experts in risk management, surgeons, anesthetists, cardiologists, internists, and nursing coordinators. The study model was focused on the perioperative management of oral anticoagulant therapy in patients who are candidates for elective surgery, especially prosthetic surgery.

The multidisciplinary group studied the process and identified four main phases of the perioperative management of oral anticoagulants:Surgical indication;Preoperative evaluation;Perioperative management;Discharge and follow-up.

Each of these phases was subsequently divided into micro-activities to identify the related possible failure modes and their potential consequences.

The RPN was calculated for each failure mode on a predetermined score for each of the individual variables: severity (S), occurrence (O), and detectability (D) of the event, as shown in Table 1.

Based on the RPN value, four categories of intervention priority have been identified: RPN > 30 “very high”; 20 < RPN < 29 “high”; 10 < RPN < 19 “medium”; RPN < 10 “low”. Therefore, a priority master list was obtained, synthesizing the characteristics considered in the FMEA.

This pilot project was applied to the perioperative management of patients treated with OA undergoing elective surgery at the Nuova Itor Hospital, a contract clinic in Rome, whose activity volumes in 2021 were 3571 surgical procedures. Among them, 532 involved prosthetic surgeries, in detail: 214 hip arthroplasties, 261 knee arthroplasties, 21 shoulder arthroplasties, and 24 prosthesis revisions. In addition, 12 surgical procedures were performed for proximal femur fractures. The main features of patients undergoing prosthetic surgery are listed in Table 2.

## 3. Results

Seventeen failure modes were identified in the entire perioperative period, of which more than half were related to the preoperative assessment. Table 3 shows the priority master list derived from the application of the FMEA to the entire perioperative process.

As shown in Table 3, the high-priority effect of the identified errors is the increased risk of bleeding, either in relation to surgical procedure or locoregional anesthesia, in the latter case resulting in an increased risk of spinal/epidural hematoma.

## 4. Discussion

The central objective of the correct OA therapy management in the perioperative period is to ensure a balanced thrombotic-hemorrhagic risk to patient undergoing elective surgery. 

According to the American College of Chest Physicians [13], high thrombotic risks include: Patients with a mechanical heart valve: any mechanical mitral valve; caged ball or tilting disk valve in mitral/aortic position; recent (<6 months) stroke or TIA;Patients with atrial fibrillation: CHADS2 score of 5 or 6; recent (<3 months) stroke or TIA; rheumatic valvular heart disease;Patients with venous thromboembolism: recent (<3 months) VTE; severe thrombophilia; deficiency of protein C, protein S, or antithrombin; antiphospholipid antibodies; multiple thrombophilia.

Meanwhile, bleeding risk can be categorized into patient and procedural-related hemorrhagic factors. Regarding the subject, important factors are the family and personal history of acquired (particularly associated with surgery) or hereditary bleeding disorders, comorbidities, concomitant medications, and laboratory tests such as platelet count, prothrombin time (PT), activated partial thromboplastin time (aPTT), renal and liver function should be considered [14]. On the other hand, both surgical or anesthesiologic risk will define the eventual timing of OA interruption and resumption [15,16]. Major surgery with extensive tissue injury, cancer surgery, major orthopedic surgery, reconstructive plastic surgery, bowel resection, cardiac, intracranial, or spinal surgery are high bleeding risk procedures [17]. As concerns VKAs, in the case of surgery, the INR target is ≤1.4 [13]. Therefore, in selected cases, AVK suspension and its temporary replacement by heparin, known as “bridging therapy”, may prove necessary before and/or after surgery [18]. However, bridging therapy is not recommended for DOACs [19]. In this case, the timing of suspension only depends on the renal function of the patients and the bleeding risk of the procedure. This is because DOACs are characterized by a rapid onset and a short half-life, so that their suspension before surgery (pharmacokinetic strategy) should reduce drug concentration according to each bleeding risk. It should be noted that the elimination of DOACs, especially Dabigatran Etexilate, is mainly dependent on renal function. Therefore, for the pharmacokinetic strategy to be effective, it is necessary to evaluate the exact timing of the last dose according to the patient creatinine clearance evaluated before the procedure [20,21].

As stated so far, the present study highlighted that the most critical part of the perioperative management of patients treated with OA is the pre-operative stage.

In detail, the failure modes that are connected with a higher level of concern, and that should be prioritized, are: the incorrect timing of OA suspension, because of incomplete evaluation of the patient bleeding risk (i.e., lack of systematic evaluation of glomerular filtration rate in the preoperative phase), the incorrect scheduling of the surgical intervention, and the incorrect or incomplete assessment of surgical procedure bleeding risk. Regarding the hemorrhagic risk of the procedure, it should be emphasized that it must be considered both at surgical and anesthetic sites for the risk of spinal hematoma from inappropriate management of DOACs [22,23].

Moreover, anticipation or postponement of the surgical procedure with respect to the scheduled date will require a rearrangement of the anticoagulant withdrawal plan. Otherwise, the patient will be exposed to an increased risk of bleeding if the procedure is anticipated, and to increased thrombotic risk if the procedure is postponed, especially regarding DOACs, since for VKAs it is sufficient to continue with bridging therapy.

At present, scientific evidence recommends bridging therapy for patients treated with AVK who are at high risk of thromboembolic events. In contrast, EBPM use is not indicated in patients with intermediate and low risk, although an evaluation that considers the thrombotic and hemorrhagic profile in the individual patient must always be carried out [24,25]. Therefore, the accurate planning of bridging therapy requires a careful evaluation of the thromboembolic risk of the patient at the moment of surgical planning and its definition during the pre-surgical phase, to avoid the increased hemorrhagic risk of the surgical procedure.

Likewise, an incorrect assessment of the hemorrhagic risk of the procedure and the patient’s thrombotic risk determine an inadequate suspension of anticoagulant therapy, exposing the patient to the risk of thromboembolic [26] or hemorrhagic events either at the surgical or anesthesiologic site [27].

At the end of the analysis carried out, the working group developed improvement actions and related monitoring methods of the introduced corrective measures, as summarized in Table 4.

## 5. Conclusions

The perioperative management of anticoagulant therapy represents, in daily clinical practice, a topic that raises several concerns; as already mentioned, patients on anticoagulant therapy are often elderly patients, with multiple prothrombotic or hemorrhagic risk factors and multiple comorbidities that affect hemostatic function. In this scenario, preventing the risks associated with the perioperative phase represents a challenge for clinicians.

Risk management consists of several phases represented by risk assessment, control, review, and monitoring [28,29,30,31]. This approach allows the identification of different sources of variability that should be kept under control for a correct management of the protocols, which must be designed in a very clear way in order to minimize errors and to limit the side effects on results. The FMEA can be an excellent and practical starting point for clinicians to identify potential causes of failures in a clinical process, as a first step to implement the current management and control of specific procedures, protocols, and clinical pathways.

The FMEA application to the perioperative management of anticoagulant drugs is a useful tool which is able to highlight the potential solutions to the critical steps of the existing procedures. In our case, following the outcome of the FMEA, the hospital procedure was updated, introducing new models and forms for the appropriate interruption and resumption of drug therapy. 

This pilot study confirms that FMEA is a valuable prospective analytical method that can be applied to most processes in healthcare. Indeed, it is well known in the literature that FMEA can be successfully applied to evaluate the safety of existing procedures, process changes, and evaluate the introduction of new processes. There are widespread literature studies that have used FMEA to evaluate the impact of different situations such as drug shortages and to select the best choice between two alternatives; or the procedures for administering therapies (chemotherapy, radiotherapy, transfusion, pharmacological); communication and patient handoff; as well as processes in particularly delicate areas (clinical laboratory; intensive care unit) [32,33]. Indeed, current healthcare systems are encouraged to focus more on safer systems rather than safer individuals. FMEA focuses on systems and investigates system failures and not individual errors, which makes this method more suitable for healthcare process analysis.

However, the work carried out so far represents only a part of the entire development of the task of the working group. To translate the identified actions into practice, it will be necessary to train and raise the awareness of the health personnel in charge, and adequately allocate time and resources for the control activities in order to be able to monitor applications and measure outcomes.

## 6. Study Limitations

As widely known, teams of Patient Safety experts use FMEA to evaluate processes to identify possible “failures” within the clinical-diagnostic-care pathways and to prevent them, introducing changes or barriers to proactively stop the occurrence of adverse events [34]. This emphasis on prevention has the purpose of reducing the risk of harm to both patients and healthcare staff. In this sense, the FMEA is particularly useful for evaluating a new process before its implementation and for evaluating the impact of a proposed change on an existing process.

However, the FMEA has, like any instrument, some disadvantages/limitations that should be kept in mind during the different phases of its execution, to apply the results critically, namely: (a)The need for a strong initial investment in time and resources;(b)Close dependence on the contribution and experience of the group, since the same analysis performed by different working groups can lead to slightly different results;(c)The trade-off between completeness and practicality (if every problem ended up on an FMEA sheet, the analysis could become very long and complex) [35].

## Figures and Tables

**Table 1 ijerph-19-16430-t001:** RPN variables, labeling, and relations to their scores.

Score	Severity	Occurrence	Detectability
1	No harm	Very low (1:10,000)	Very high (9:10)
2	Mild harm	Low (1:5000)	High (7:10)
3	Moderate harm	Moderate (1:200)	Medium (5:10)
4	Severe harm	High (1:100)	Low (2:10)
5	Death	Very high (1:20)	Very low (<1:10)

**Table 2 ijerph-19-16430-t002:** Main characteristics of the patient undergoing an elective prosthetic surgery at Nuova Itor Hospital in 2021.

Sex (M, %)	36.40%
Age (y)	70.36
Days of hospitalization (d)	6.45

**Table 3 ijerph-19-16430-t003:** Proposed master list as resulted from the failure modes of the FMEA application to perioperative management of OA therapy.

Function	Potential Failure Mode(s)	Potential Effect(s)	RPN
Patient’s thrombotic risk assessment	Wrong DOAC timing suspension	Increased hemorrhagic risk of the surgical procedure	35
Patient’s thrombotic risk assessment	Wrong VKAs suspension—wrong “bridging therapy”	Increased hemorrhagic risk of the surgical procedure	25
Surgery planning	Anticipation of surgery without rescheduling DOACs management	Increased hemorrhagic risk of surgical procedure	24
Surgery hemorrhagic risk assessment	Wrong hemorrhagic risk assessment of the surgical procedure	Increased thromboembolic risk	21
Anesthesiologic hemorragic risk assessment	Wrong DOAC timing suspension	Increased hemorrhagic risk at the locoregional anesthetic site	20
Surgery planning	Anticipation of surgery without rescheduling OA management	Increased hemorrhagic risk at the locoregional anesthetic site	20
Application of the scheduled OA therapy suspension plan	Wrong postoperative management of OA therapy	Increased hemorrhagic risk of the surgical procedure	18.75
Surgery hemorrhagic risk assessment	Wrong hemorrhagic risk assessment of the surgical procedure	Increased hemorrhagic risk of surgical procedure	18
Surgery planning	Delayed surgery without rescheduling DOACs management	Increased thromboembolic risk	17.5
Application of the scheduled OA therapy suspension plan	Wrong postoperative management of OA therapy	Increased thromboembolic risk	15
Patient’s thrombotic risk assessment	Wrong VKAs suspension: wrong “bridging therapy”	Increased hemorrhagic risk at the locoregional anesthetic site	15
Surgery planning	Anticipation of surgery without reprogramming VKAs management	Increased hemorrhagic risk of the surgical procedure	14
Patient’s thrombotic risk assessment	Wrong OA therapy suspension	Increased thromboembolic risk	14
Patient’s thrombotic risk assessment Definition of the scheduled OA therapy suspension plan	Wrong OA therapy timing suspension or wrong sharing of the preoperative management plan in the multidisciplinary team	Need to postpone the scheduled surgery for the detected error	9.37
Surgery planning	Anticipation of surgery without reprogramming DOACs management	Need to postpone the scheduled surgery for the detected error	7.5

**Table 4 ijerph-19-16430-t004:** Proposed improvement plan to face failure modes with a higher level of concern and priority from FMEA working group.

Process/Activity	Improvement Action	Monitoring
Wrong DOAC timing suspension: increased hemorrhagic risk of the surgical procedure	Standardize on EBM (Evidence Based Medicine) basis DOAC suspension according to the surgical procedure risk and renal function assessed in the pre-hospitalization phase. Discuss with multidisciplinary team involved in pre-hospitalization Draft DOAC suspension forms, according to each pharmacokinetics, as support for healthcare professionals	Periodic verification of the correct compilation of the forms adopted and the involvement of the multidisciplinary team reported in patient medical records
Wrong VKAs suspension—wrong “bridging therapy”: Increased hemorrhagic risk of the surgical procedure	Standardize TAO suspension: Discuss with multidisciplinary team involved in pre-hospitalization Share a thrombosis prophylaxis protocol, drafted according to EBM basis; Share and use TAO management remainder form during the pre-hospitalization: Suspension scheme and switch to “bridging therapy”	Periodic verification of the correct compilation of the forms adopted and the involvement of the multidisciplinary team reported in patient medical records
Anticipation of surgery without rescheduling OA management: Increased hemorrhagic risk at both locoregional anesthetic and surgical sites	Any rescheduling of the intervention in advance or postponement of the same must be systematically shared by the surgeon with a multidisciplinary pre-hospitalization team with re-evaluation of the management of anticoagulants.	Periodic verification of the involvement of the multidisciplinary team reported in patient medical records
Wrong hemorrhagic risk assessment of the surgical procedure: Increased hemorrhagic risk of the surgical procedure	o Avoid synthetic abbreviations of the planned intervention in the pre-hospitalization documentation and waiting lists; o Always involve the orthopedist/surgeon during the pre-hospitalization phase and within the multidisciplinary team.	Periodic verification of the involvement of the multidisciplinary team reported in patient medical records
Wrong postoperative management of OA therapy	o Identify by the multidisciplinary team, on EBM basis, the resumption timing of VKA and DOAC, which can only be assessed in postoperative phase (depending on the clinical conditions, as well as postoperative laboratory data). o Share with the healthcare staff the use of the specific VKA and DOAC management forms developed ad hoc by the FMEA group; o Report clear information in the discharge letter and provide a reminder to the patient, for his general practitioner or other physicians if addressed to another facility (e.g., for rehabilitation).	Periodic verification of the involvement of the multidisciplinary team reported, correct completion of the discharge letter and delivery of the remainder forms, a copy of which is kept in patient medical records.

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
