# Peer review of "Proactive Risk Assessment through Failure Mode and Effect Analysis (FMEA) for Perioperative Management Model of Oral Anticoagulant Therapy: A Pilot Project"

_ijerph, 2022, doi:10.3390/ijerph192416430_

Round 1

Reviewer 1 Report

Content suggestions:

  1. The authors mentioned Table 3 and Table 4 in the text. However, even when describing Table 3 in the manuscript, the table below is entitled as Table 4. I suppose that it should be assigned as Table 3 also in the text and that Table 4 was not planned to be prepared.

  2. Can the authors characterize the statistical methods used in the study ?

  3. Can the authors provide more information about the comorbidities of the patients ? What about allergies as further adverse effect of the treatment – do the auhots have this information ?

The manuscript can be published after minor corrections made on the basis of the content suggestions.

Author Response

26th October, 2022
Dear Editor
We thank the Referees for their interest in our work and for helpful comments that will greatly improve the manuscript (Manuscript ID: ijerph-1983861). The Referees have brought up some good points and we appreciate the opportunity to increase the quality of our study report.

Reviewer 1:

Thanks for the comment. We corrected the errors.

1. The authors mentioned Table 3 and Table 4 in the text. However, even when describing Table 3 in the manuscript, the table below is entitled as Table 4. I suppose that it should be assigned as Table 3 also in the text and that Table 4 was not planned to be prepared.

2. Can the authors characterize the statistical methods used in the study?

The study involved the application of a methodology, the "FMEA", well known by the scientific community for the analysis of processes in the health field. The Risk Priority Number (RPN) was calculated for each failure mode on a predetermined score for all the individual variables, as described in the Table 1 of the manuscript.

3. Can the authors provide more information about the comorbidities of the patients ? What about allergies as further adverse effect of the treatment – do the authors have this information ?

Although this information is present in medical records of the patients involved in the study, the design of the present study is not that of a clinical trial, therefore these clinical aspects do not influence the results of the analysis.

We really appreciated your help in improving our work.

Please do not hesitate to contact me for any further questions. Dr. Giuseppe Bertozzi
Department of Clinical and Experimental Medicine, University of Foggia, Foggia, 71100, Italy giuseppe.bertozzi@unifg.it

tel. +39 3401495648

Reviewer 2 Report

Dear authors,

thank you very much for conducting the present study. 

Although the topic seems important and interesting, this manuscript needs a  thorough revision. 

In general you should avoid the "bloomy" writing and give more concrete information important to the study. 

The design of your study is not clear, neither the "clinical" evaluation. You should be more concrete how and what you evaluated in the clinical setting.

It is to clear what kind of data you collected to proof your model. this comes back in the results section.  

The information on your patients is poor. This needs much more detail. 

Regarding the ethical aspects, I miss the IRB number and a statement about the consenting. 

Regarding the data, table 3 (which seems to be the most relevant is missing). 

The presented results itself, are without any substance. 

The discussion is partially a repetition of the introduction and gives no concrete results or the interpretation of them. There is no comparison to the literature. What is the advantage of your method above scoring systems like Has-Bled or Chads scores? Why are your results new? Bad timing is the most relevant, but certainly best known factor...

The conclusion, again, does not give the final statement but more a repetition of what's been said. 

Kindly perform a thoroughly revision. 

Author Response

26th October, 2022
Dear Editor
We thank the Referees for their interest in our work and for helpful comments that will greatly improve the manuscript (Manuscript ID: ijerph-1983861). The Referees have brought up some good points and we appreciate the opportunity to increase the quality of our study report.

Reviewer 2:

Dear authors,

thank you very much for conducting the present study. Although the topic seems important and interesting, this manuscript needs a thorough revision.

In general you should avoid the "bloomy" writing and give more concrete information important to the study.

Thanks for the advice. We have extensively revised the manuscript.

The design of your study is not clear, neither the "clinical" evaluation. You should be more concrete how and what you evaluated in the clinical setting. It is to clear what kind of data you collected to proof your model. this comes back in the results section. The information on your patients is poor. This needs much more detail

The design of the present study provided the application of a proactive methodology, the "FMEA", well known by the scientific community for the analysis of processes in the health field. Therefore, as the present work is not a clinical trial, although clinical information of the patients was available to the authors, it is not relevant for the analysis. Otherwise, the activities volumes have been reported to explain the expertise of the working group which represents a limitation of the FMEA method, as stated in the paragraph n.6 of the manuscript.

Regarding the ethical aspects, I miss the IRB number and a statement about the consenting.

As it is not a clinical study, patients who underwent a surgical procedure in the Nuova Itor Hospital in 2021 were required to consent to the use of their anonymised data in compliance with Italian privacy legislation.

Regarding the data, table 3 (which seems to be the most relevant is missing).

Thanks for the comment. We corrected the error.

The presented results itself, are without any substance. The discussion is partially a repetition of the introduction and gives no concrete results or the interpretation of them. There is no comparison to the literature. What is the advantage of your method above scoring systems like Has-Bled or Chads scores? Why are your results new? Bad timing is the most relevant, but certainly best known factor... The conclusion, again, does not give the final statement but more a repetition of what's been said.

Kindly perform a thoroughly revision.

To facilitate understanding of the contribution provided by the present study, it has been included the table 4, which reports the improvement plan of the perioperative management of anticoagulants.

The FMEA method is not comparable with the Has-Bled or CHA2DS2-VASc score. In fact, the FMEA is used for the systematic analysis of risk in complex systems such as health care (in the present study it was applied to the management of perioperative anticoagulant therapy, but it can be applied in any field). Its purpose, therefore, is to recognize, limit and remedy potential weaknesses and thus avoid errors. In contrast, Has-Bled or CHA2DS2-VASc clinical scores are applied in a purely clinical setting to assess a patient's bleeding risk.

Therefore, thanks to your notes, we have improved the manuscript.

We really appreciated your help in improving our work.

Please do not hesitate to contact me for any further questions. Dr. Giuseppe Bertozzi
Department of Clinical and Experimental Medicine, University of Foggia, Foggia, 71100, Italy giuseppe.bertozzi@unifg.it

tel. +39 3401495648